# SnAKe: Bayesian Optimization via Pathwise Exploration

**Jose Pablo Folch** *
Imperial College London
London, UK

**Shiqiang Zhang**
Imperial College London
London, UK

**Robert M Lee**
BASF SE
Ludwigshen, Germany

**Behrang Shafei**
BASF SE
Ludwigshafen, Germany

**David Walz**
BASF SE
Ludwigshafen, Germany

**Calvin Tsay**
Imperial College London
London, UK

**Mark van der Wilk**
Imperial College London
London, UK

**Ruth Misener**
Imperial College London
London, UK

## Abstract

Bayesian Optimization is a very effective tool for optimizing expensive black-box functions. Inspired by applications developing and characterizing reaction chemistry using droplet microfluidic reactors, we consider a novel setting where the expense of evaluating the function can increase significantly when making large input changes between iterations. We further assume we are working asynchronously, meaning we have to select new queries before evaluating previous experiments. This paper investigates the problem and introduces 'Sequential Bayesian Optimization via Adaptive Connecting Samples' (SnAKe), which provides a solution by considering large batches of queries and preemptively building optimization paths that minimize input costs. We investigate some convergence properties and empirically show that the algorithm is able to achieve regret similar to classical Bayesian Optimization algorithms in both synchronous and asynchronous settings, while reducing input costs significantly. We show the method is robust to the choice of its single hyper-parameter and provide a parameter-free alternative.

## 1 Introduction

We introduce a method for black-box optimization that keeps step-wise input variations as small as possible. A black-box function is expensive to evaluate (with respect to time or resources), and we do not have access to gradients. Classically, black-box optimization finds an optimum by sequentially querying the function. We study a variation of this problem, with two important differences. First, we introduce the idea that large changes in inputs, between iterations, cause the function to become more expensive to evaluate. Second, we do not assume observations are available immediately: delay between querying the function and getting a result leads to asynchronous decision making.

As a motivating example, consider a droplet microfluidic reactor [49] (see Figure 6 in Appendix). In such a reactor, we can quickly pump in chemicals, expose them to certain conditions, and collect the results of our experiments as they exit. However, large changes in temperature mean that the reactor is no longer in steady-state, and this makes the evaluations unreliable until the system stabilizes.

---

*Corresponding author: `jose.folch16@imperial.ac.uk`

36th Conference on Neural Information Processing Systems (NeurIPS 2022).

Smaller changes can maintain a "quasi-steady state", as the system is easier to stabilize. Further, we have to wait for droplets to exit the reactor before obtaining observations.

More broadly, consider optimization requiring spatially continuous exploration. Black-box optimization has been investigated for optimal placement of pollution sensors [17], and machine learning methods for detecting the sources of pollution have been developed [30]. Mobile pollution readers, such as those used by AirQo [`https://www.airqo.net/about`], would face similar challenges if black-box optimizers were used to track the pollution sources. Samaniego et al. [41] show a more concrete example where autonomous vehicles use black-box optimizers to monitor and track water contamination sources in the Ypacarai lake in Paraguay.

Classical Bayesian Optimization (BO) [20, 42] provides effective solutions to black-box optimization; however, physics-based limitations are usually not taken into account [50]. Typically BO follows a myopic approach, in that BO chooses the next query based only on the current state of the surrogate model. Classical BO reduces uncertainty in unexplored areas and then returns to promising areas with no regard for the distance between consecutive query points. This means that BO will incur very high *input costs*.

However, having zero changes in input space is obviously not a good solution. After all we want to explore the search space to find the optimum point. We seek an algorithm that preserves the essence of Bayesian Optimization. Consider a scenario where we know a large number of inputs we want to query. In this case, we could simply order the queries to attain the smallest input cost. A good solution would simply require selecting a large number of queries, creating an ordering, and then following the *path* defined by the ordering. Once we start obtaining new information, we could update our beliefs and update our optimization path.

This paper proposes *Sequential Bayesian Optimization via Adaptive (K)Connecting Samples* (SnAKe). Just as the snake grows by carefully eating items in the classic arcade game, a SnAKe optimization path grows from carefully adding queries to the evaluation path.

## 2   Related Work

'Process-constrained Batch Bayesian Optimization' [51], is the closest analog to our setting of BO with input costs. Vellanki et al. [51] navigate the physical limitations in changing the input space by *fixing* the complicated inputs for every batch. Rather than fixing the difficult-to-change inputs, we *penalize* large input variations in line with the costs (time and resources) of changing conditions. SnAKe decides when to make expensive input changes. Recently, Ramesh et al. [39] consider a similar cost setting to ours, inspired by applications to wind energy systems. Waldron et al. [52] compare the use of transient variable ramps (small input changes in a pre-determined manner) against a full steady-state design of experiments approach for learning parameters of chemical kinetic models. They show that the transient approaches give less precise estimates, but much faster. SnAKe combines the best of both worlds, automatically designing experiments while keeping input changes small.

'Cost-aware Bayesian Optimization' [28, 31, 44] optimizes regret with respect to cost per iteration. These costs, which are fixed throughout the BO process, arise when some regions of the input domain are more expensive to evaluate than others. Similar ideas appear in Multi-fidelity BO [21, 37] where cheap approximations for the objective function are used. Bogunovic et al. [6] combines cost-aware and level-set estimation approaches to solve environmental monitoring problems. Diverging from prior works, our setting does not assume that different regions incur different costs. Instead, SnAKe addresses the setting where the *difference* between adjacent inputs defines costs. The query order changes the cost, therefore we focus on optimization paths instead of optimization sets.

The baseline of Cost-aware BO, Expected Improvement per unit Cost (EIpu), defined as $\mathrm{EI}(x)/\mathrm{Cost}(x)$, is not directly applicable to our setting because the cost of not changing inputs, i.e., choosing $x_t = x_{t-1}$, is zero. We can instead use:

$$\gamma \mathrm{EIpu}(x) = \mathrm{EI}(x)/(\gamma + \mathrm{Cost}(x, x_{t-1})) \tag{1}$$

Three main issues: (a) Eq. (1) introduces new hyper-parameter $\gamma$ with no obvious way to choose it, (b) Eq. (1) effectively penalizes the acquisition function far away from the data, so near local optima we only penalize exploration and may over-exploit, (c) asynchronous extensions have a tendency to encourage batch diversity, so we introduce a new trade-off between penalizing locally and far away. As we shall see, SnAKe shows robust results with just one hyper-parameter.

*Look ahead* Bayesian Optimization [12, 15, 26] considers possible future queries to make the current choice less myopic. We could use it to select the inputs to query, as we could simply look-ahead at what any classical BO would choose and then order the queries accordingly. However, one is only able to look ahead for a few iterations, due to the computational complexity of looking far ahead into the future [5, 54]. For example, Lee et al. [29] combines Markov Decision Processes [38], look-ahead and cost-aware ideas, but is limited by the short time horizons rollout can handle. Reinforcement Learning [48] can be used, but it requires access to a dedicated training environment. Mutný et al. [34] show how similar ideas can be used for trajectory optimization in environmental monitoring.

A computationally cheaper alternative that allows us to select many inputs to query is Batch Bayesian Optimization (BBO) [2, 14]. BBO is the setting where we are able to parallelize function evaluations, and as such we want to select multiple queries simultaneously. González et al. [15] and Jiang et al. [19] link BBO with look-ahead BO, using a Local Penalization method [14] and Expected Improvement (q-EI) [13], respectively. Unfortunately, they restrict themselves to smaller batch sizes ($q \leq 15$) due to computational expense. *Asynchronous* Bayesian Optimization [1, 22] addresses the problem of choosing queries while waiting for delayed observations.

## 3 Methods

### 3.1 Problem Set-up

We consider finding the maximum, $x^* = \arg\max_{x \in \mathcal{X}} f(x)$, of a black-box function, $f$, where $\mathcal{X}$ is a compact subset of $\mathbb{R}^d$. We assume $f$ is continuously differentiable and expensive to evaluate. We seek the optimum point while keeping the number of evaluations small. We evaluate the function sequentially, over a discrete and finite number of samples, $t = 1, ..., T$. For every query, $x_t$, we obtain a noisy observation of the objective, $y_t = f(x_t) + \eta_t$, where $\eta_t \sim \mathcal{N}(0, s^2)$ is Gaussian noise.

We assume there is delay, $t_{delay}$, between choosing a query and getting an observation. So our data-set at iteration $t$ is given by $D_t = \{(x_i, y_i) : i = 1, ..., t - t_{delay} - 1\}$. If we set $t_{delay} = 0$, we revert to classical sequential Bayesian Optimization, otherwise we are in an *asynchronous* setting.

Finally, we assume there is a *known* cost to changing the inputs to our evaluation, $\mathcal{C}(x_t, x_{t+1})$. We use simple regret, $SR_t = f(x^*) - \max_{i=1,...,t} f(x_i)$ as the performance metric. We want to minimize regret, or equivalently, maximize $f$, for the smallest possible cumulative cost, $\sum_{t=1}^{T-1} \mathcal{C}(x_t, x_{t+1})$.

We note that the problem definition is invariant to cost scaling relative to the objective, because an optimal trajectory will be defined solely by the shape of the function. Current cost-aware methods are not scale invariant, and as we shall see, SnAKe first chooses which points to query and then finds the most cost-efficient way of selecting them, making the method invariant to scaling of the cost.

As is common in Bayesian Optimization, we will model the black-box function by putting a Gaussian Process (GP) prior on $f \sim \mathcal{GP}(\mu_0, \sigma_0^2)$. Since we have Gaussian noise, the posterior, $f|D_t$ is also a GP, whose mean function, $\mu_t(\cdot)$, and covariance function, $\kappa_t(\cdot, \cdot)$, can be calculated analytically [40].

### 3.2 General Approach

For our general approach, we will seek to create a large *batch* of queries that we want to evaluate, and then *plan-ahead* a whole optimization path. This is useful for three reasons: (1) it allows us to order the queries in a way that reduces input cost. (2) It allows us to deal with any delay in getting observations, because we can pre-select future queries. (3) It remains computationally feasible to plan-ahead even for very large time horizons. We can follow this ordering or path until new information is available, after which we will update our path.

While we will provide a specific way of creating, planning and updating the paths, we note that the general ideas can be extended to suit different settings. For example, Appendix C shows how to alter SnAKe to simultaneously optimize multiple, independent, black-box functions.

### 3.3 Creating a Batch Through Thompson Sampling

We need to produce batches that are representative of the current state of the surrogate model. In addition, the method should allow for big batch sizes, since we want to produce batches as big as our

budget (which is usually much larger than the batch size most methods consider). For example, in a micro-reactor, we might be interested in batches that contain hundreds of points [49].

Kandasamy et al. [22] offers a promising solution where every point in the batch is independent. The method is based on Thompson Sampling, which uses the GP's inherent randomness to create a batch. Each batch point is chosen by drawing a realization of the GP, and optimizing it. The queries will fill out the space, and they are more likely to be on promising, and unexplored areas. It should work very well in our context given we expect our initial batch sizes to be very large, so the sample should be representative of the current state of our surrogate model.

## 3.4 Creating a Path via the Travelling Salesman Problem

After selecting a batch of queries, $\mathcal{P}_t = \{x_t^{(i)}\}_{i=1}^{\tilde{T}}$, we order them. We do this by embedding a graph into the batch, where the edge weights are the cost for changing one input to another. We then find the shortest path that visits every point, i.e., we solve the Travelling Salesman Problem (TSP) [4, 8]. Section 3.9 discusses the computational cost.

Mathematically, we define the graph $G = (V, E, W)$, with $V = \{i \in 1, ..., \tilde{T} : x_t^{(i)} \in \mathcal{P}_t\}$, $E = \{(i, j) : i, j \in 1, ..., \tilde{T}\}$, and $W = \{w_{ij} = \mathcal{C}(x_t^{(i)}, x_t^{(j)}) : (i, j) \in E\}$, where $\tilde{T}$ is the number of batch samples. We solve the TSP in $G$ to obtain our latest optimization path. A simple example would be to try to minimize the total distance travelled in input space, by selecting the Euclidean norm as cost $\mathcal{C}(x_t, x_{t+1}) = ||x_t - x_{t+1}||$.

## 3.5 Naively Updating the Optimization Path

After updating the GP with new observations, we want to use this information to update our path. We propose updating our strategy by creating a new batch of points.

At iteration $t$, the remaining budget has size $T - t$. We first propose sampling $T - t$ queries through Thompson Sampling, and then solving the Travelling Salesman Problem. However, we show this leads to the algorithm getting 'stuck' in local optima. This is because every time we re-sample, we naturally include *some exploitation* in the batch, and this exploitation will always be the next point chosen by the TSP–we will never reach the *exploration* algorithm steps. Figure 1a shows an example.

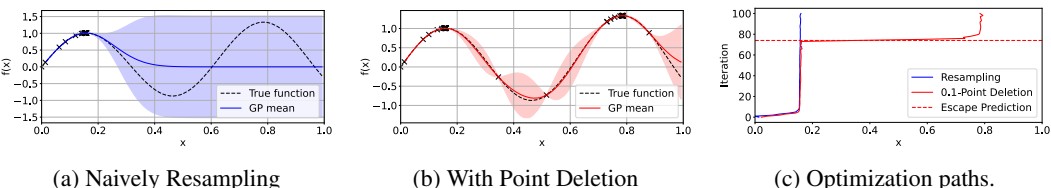

| (a) Naively Resampling | (b) With Point Deletion | (c) Optimization paths. |

Figure 1: Effect of Point Deletion for ordering-based BO. (a) and (b) show the maximization objective and the underlying surrogate model, with black crosses representing queries and observations. (c) shows the optimization paths. Naive resampling gets stuck in the local optimum ($x = 0.16$) whereas Point Deletion escapes the local solution. We also show the predicted escape iteration, $Tp$ (after estimating $p \approx 0.74$ in Appendix D.1), which accurately predicts the algorithm behavior in this simple example.

## 3.6 Escape Analysis

To try and solve the convergence problem introduced by naive resampling, we briefly analyze it. This analysis assumes we receive noise-less observations. However, all sampling can be done in the presence of noise by calculating the corresponding posterior. We note that this section will provide intuition for the next step of the algorithm, however, we are not providing rigorous theoretical justification for the step or proving any type of regret bounds. We focus on Thompson Sampling.

**Definition 3.1.** (Thompson Sample) We say $x_t^{(i)}$ is Thompson Sampled, and we write $x_t^{(i)} \sim \tau_t$, if:

$$x_t^{(i)} = \arg\max_{x \in \mathcal{X}} f_t^{(i)}(x), \quad \text{where} \quad f_t^{(i)}(\cdot) \sim \mathcal{GP}(\mu_t, \kappa_t | D_t)$$

Note that the sampling distribution $\mathcal{GP}(\mu_t, \kappa_t | D_t)$ changes at each iteration.

One particular concern, diagrammed in Figure 1a, is the possibility of the method being stuck in a certain area. Let $B_\delta(a)$ be a Euclidean $\delta$-ball centered at $a$, and assume that $x_{t-1} \in B_\delta(a)$. We further assume two more things: (a) we resample the batch at every iteration, and (b) the TSP solution, or any approximation, will always choose a point in $B_\delta(a)$ to be the next step, if there are any. These assumptions represent the worst-case scenario for trying to escape a local optima.

**Definition 3.2.** (Non-escape Probability) We define the non-escape probability, $p_t$, at iteration $t$, as the probability of a Thompson Sample falling into $B_\delta(a)$. That is, for $x_t^{(i)} \sim \tau_t$:

$$p_t = \mathbb{P}(x_t^{(i)} \in B_\delta(a)) = \mathbb{P}(||x_t^{(i)} - a|| \leq \delta) \tag{2}$$

Let $\mathcal{P}_t = (x_t^{(1)}, ..., x_t^{(T-t)}) \sim \tau_t$ $i.i.d.$. Of particular interest to us, is the number of 'non-escapes' in the sample, $N_t = |\mathcal{P}_t \cap B_\delta(a)|$. We are guaranteed to escape if no sample falls in $B_\delta(a)$, so we say we have *fully* escaped if $x_t^{(i)} \notin B_\delta(a)$ $\forall i$.

**Remark 3.3.** Note that $N_t \sim$ Binomial$(T - t, p_t)$ as all the samples are mutually independent, therefore the probability of fully escaping is $\mathbb{P}(N_t = 0) = (1 - p_t)^{T-t}$. This means we can only expect to fully escape if $p_t$ is very small.

Therefore we are interested in the behavior of $p_t$ as we gain more information about $f$ in $B_\delta(a)$. We now consider the circumstances under which $p_t$ becomes very small:

**Areas without stationary points:** Consider the case when our objective $f$ does not contain a stationary point in $B_\delta(a)$. Then the maximum of $f$ on the closure of the ball, $\bar{B}_\delta(a)$, must lie on the boundary of the ball. In particular, if we assume that our GP model has no error in $B_\delta(a)$, and assuming continuity of sample paths, then the non-escape probability must be zero, $p_t = 0$.

Intuitively, the area itself contains enough information to ensure, with complete certainty, that the global optimum does not lie in the area. We hope that, as we collect information inside areas without stationary points, $p_t \to 0$, and we will eventually leave them with small probability of returning. Appendix D.2 shows an example of this happening very fast.

**Areas with stationary points:** Areas with stationary points pose a much bigger problem. We will restrict our arguments to local maxima, since this is where we have observed the problem. Assume that $f$ has a local optimum in $B_\delta(a)$, higher than any other we have observed before. In this case, any sample taken from a Gaussian Process with no error in $B_\delta(a)$ will have a local maximum inside $B_\delta(a)$, and therefore it is possible that this local maximum is the global solution.

As we increase the information inside the area, $p_t$ is not guaranteed go down to zero. This makes intuitive sense; the only way of knowing if a local optimum is not a global optimum is by sampling away from it–therefore with limited information we will allocate a certain probability to the global optimum being inside $B_\delta(a)$. We include a clear example where $p_t \to p > 0$ in Appendix D.1.

Sampling consistently in a promising area is not necessarily a bad thing, indeed we want to exploit near possible global optimum candidates. However, the question then becomes, how long will it take us to leave a local optimum? Recall the probability of fully escaping is $(1 - p_t)^{T-t}$, and therefore it will be increasing as $t$ increases, even if $p_t$ is (almost) constant.

**Remark 3.4.** Assume that $p_t \to p > 0$, and that we have a high escape probability after $t_e$ iterations, i.e., $(1-p)^{T-t_e}$ is large, leaving us $T-t_e$ iterations to explore the rest of the space. If we increase our budget from $T$ to $T'$, we will not have a high probability of escape until $(1-p)^{T'-t} = (1-p)^{T-t_e}$, i.e., $t = T' - T + t_e$. This leaves us with $T' - (T' - T + t_e) = T - t_e$ iterations to explore the rest of the space. Note that this is independent of $T'$, meaning that increasing our budget *does not* increase our budget after leaving $B_\delta(a)$! Rather, it only means we will be stuck in $B_\delta(a)$ for a longer time. This is very concerning as the method will be very myopic; if it finds a local optimum, it is likely that it will spend a very large amount of the budget exploiting it.

### 3.7 Escaping with $\epsilon$-Point Deletion

Intuitively, the convergence problem stems from the constant resampling, because in every iteration resampling is reintroducing exploitative points. We seek a way of making sure that the samples

used by the algorithm to create paths take into account the previously evaluated designs. Penalizing the samples via distances to previous queries seems like a simple solution, e.g., similar to local penalization [14]. However, penalization introduces the need to carefully tune the strength of the penalization so that we can still exploit optima and remain cost-effective – the parameterization is nontrivial.

Around a local maximum, if we assume that $p_t \to p > 0$, then we can interpret $p$ as the probability that the global maximum lies on the ball $B_\delta(a)$, as such, we focus on a method that has the property of exploiting the promising area for $Tp$ iterations (that is, the total budget weighted by the probability). To achieve this, we propose creating a total of $T$ samples, and then deleting batch points that are similar to previously-explored points (i.e., we remove excess exploitation introduced by resampling). Algorithm 1, which we term $\epsilon$-Point Deletion, still allows us to exploit local optima if we sample many points near them, however, it should eventually move on.

Two points to note: (a) Point Deletion uses the Euclidean norm, and it is independent of the cost function. This is because we are trying to escape local minima of simple regret. (b) After the deterministic removals, it is likely the batch size is still larger than our remaining budget, so we balance it by randomly removing points from the batch.

---

**Algorithm 1** $\epsilon$-Point Deletion

**input:** New proposed batch $\mathcal{P}_t$ (size $T$), set of already queried points $Q_t$ (size $t$), and deletion distance $\epsilon$
**for** $x \in Q_t$ **do**
    $\tilde{d} \leftarrow \min_{x' \in \mathcal{P}_t} ||x - x'||$
    **if** $\tilde{d} < \epsilon$ **then**
        `# find the closest point to the query` $x$ `in the new batch`
        $\tilde{x} \leftarrow \arg\min_{x' \in \mathcal{P}_t} ||x - x'||$
    **else**
        `# else pick a random sample`
        $\tilde{x} \leftarrow \text{Random}(\mathcal{P}_t)$
    **end if**
    `# remove said point from the batch`
    $\mathcal{P}_t \leftarrow \mathcal{P}_t \setminus \{\tilde{x}\}$
**end for**
**output:** A batch $\mathcal{P}_t$ (size $T - t$)

---

$\epsilon$-Point Deletion allows us to escape local optima by directly increasing the probability of fully escaping without changing $p_t$. Let $q_t$ be the number of previously queried points inside the ball, and set $\epsilon \geq 2\delta$. Indeed, it follows that we will escape if $N_t \leq q_t$, (as $q_t$ samples are guaranteed to be deleted due to previously queried points) which is much better than requiring $N_t = 0$.

Due to over-sampling, the expected number of 'non-escapes' is given by $\mathbb{E}[N_t] = p_t T$. Around a local maximum, if we assume that $p_t \to p > 0$, and hence, $p_t T \approx pT$, then we can reasonably expect an escape when $q_t = pT$, that is, we will exploit the local maximum for approximately $pT$ iterations. This time we leave $T - q_t = T(1 - p)$ extra iterations to explore the remaining space! Increasing the budget will benefit *both* the exploitation and the exploration instead of only the former (in contrast with Remark 3.4).

Figure 1 gives an empirical example where we use Point Deletion, with $\epsilon = 0.1$, to escape a local optimum. We observe the expected behavior from our brief analysis. For Point Deletion, we calculate the escape prediction as $pT \approx 74$, using $\hat{p} \approx 0.74$, which we estimated in Appendix D.1. We can see that without Point Deletion, we remain stuck in the first local optimum.

## 3.8 SnAKe

Algorithm 2 which we dub 'Sequential Bayesian Optimization via Adaptive Connecting Samples' (SnAKe), combines the ideas of previous sections. Figure 2 diagrams the most important steps of SnAKe. Section 4 develops an effective, parameter-free alternative to the choice of $\epsilon$.

Note there is no requirement for data to be available immediately following querying. If $t_{delay} > 0$, we can simply stick to the latest path. It works without modification on the asynchronous setting. This is vital since we were inspired by chemical experiment design which can exhibit asynchronicity. We also note that the problem is invariant to cost scaling, as the batch creation and point deletion steps are independent of the cost, and the solution to the Travelling Salesman Problem is also scale-invariant.

---

**Algorithm 2** SnAKe

**input**: Optimization Budget, $T$. Deletion constant, $\epsilon$.
**begin**: Create initial batch, $\mathcal{P}_0$, uniformly. Choose starting point $x_0$. $Q_0 \leftarrow \{x_0\}$. Create initial path, $S_0$, by solving TSP on $\mathcal{P}_0$.
**for** $t = 1, 2, 3, ..., T$ **do**
    Check if any running evaluations are finished
    **if** there are new observations **then**
        Update surrogate model
        Create batch of size $T$ using Thompson Sampling, $\mathcal{P}_{t-1}$
        $\mathcal{P}_{t-1} \leftarrow \epsilon\text{-Point Deletion}(\mathcal{P}_{t-1}, Q_{t-1})$
        $\tilde{S} \leftarrow \text{TSP}(\mathcal{P}_t, \text{source} = x_{t-1}) \setminus \{x_{t-1}\}$
        $S \leftarrow Q_{t-1} \cup \tilde{S}$
    **end if**
    Choose next query point from schedule: $x_t \leftarrow S_t$
    $Q_t \leftarrow Q_{t-1} \cup \{x_t\}$
    Evaluate $f(x_t)$
**end for**

---

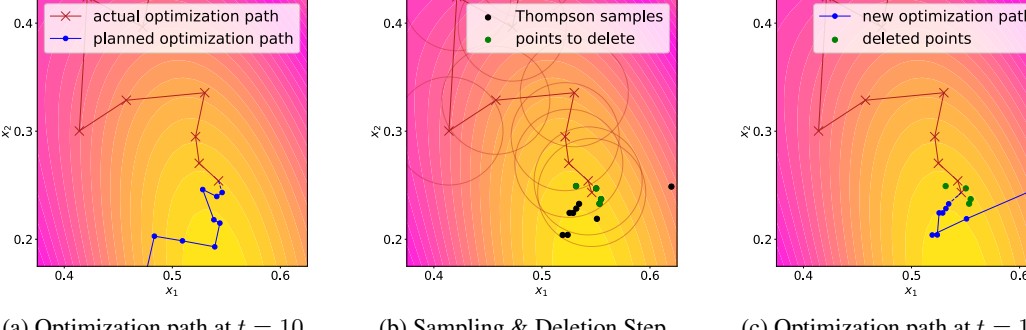

(a) Optimization path at $t = 10$      (b) Sampling & Deletion Step      (c) Optimization path at $t = 11$

Figure 2: Graphical example of SnAKe behavior in a full iteration if new information is available. The underlying function is Branin 2D. The feasible set is $[0, 1]^2$, so we do not see all the samples or the complete path with this zoomed-in view. (a) The red line shows the already-queried path. The blue path shows our future plans. (b) For each query, we can see the $\epsilon$-ball under which the deletion step is deterministic. We plot the Thompson Samples as dots, the accepted ones in black and the deleted ones in green. (c) The new path is in blue and the points ignored (due to Point Deletion) are in green. Note the higher concentration of samples in what the model considers a promising area.

## 3.9 Computational Considerations

We use Wilson et al. [53] to efficiently sample the GP and optimize the samples using Adam [23]. The optimization step may be expensive, but it is highly parallelizable if required. We heuristically solve the Travelling Salesman Problem with Simulated Annealing [24] and use an adaptive grid to reduce the number of samples away from our current input. The grid consists of the closest $N_l$ samples to our current input; the remaining samples are assigned to one of $N_g$ points in a global grid. Coarsifying the placement of the far-away points effectively ignores detailed paths far away from our current input. These far-away points are not important to the problem, as long as we resample frequently enough. We introduce two hyper-parameters $(N_l, N_g)$, but they should not impact the solutions assuming $N_l$ is reasonably larger than any possible delay. With these modifications, we were able to comfortably run all experiments in a CPU (2.5 GHz Quad-Core Intel Core i7), where SnAKe shared a wall-time similar to Local Penalization methods. Appendix E.1 gives more details.

# 4 Experimental Results

For all experimental results we report the mean and the standard deviation over 25 experimental runs. We give the full implementation details and results in Appendix E and F respectively. Classical BO methods are implemented using BoTorch [3] and GPyTorch [11]. The code to replicate all results is available online at `https://github.com/cog-imperial/SnAKe`.

In all experiments, we examine SnAKe for $\epsilon = 0, 0.1$, and $1$. We further introduce a parameter-free alternative by adaptively selecting $\epsilon$ to be the smallest length scale from the GP's kernel, and denote it $\ell$-SnAKe. SnAKe proves to be robust to non-zero choices of $\epsilon$. We conjecture this happens because of Thompson Sampling's exploitative nature [35], so all excess exploitation is concentrated on a very small area. For $\epsilon = 0$ we observe very low cost, at the expense of some regret.

Across all experiments we can see how SnAKe can traverse the search space effectively, constantly achieving good regret at low cost. While other cost-aware methods may also achieve good performance, they are very inconsistent, suggesting they require careful tuning. SnAKe achieves good, robust results with a single hyper-parameter. We do highlight that SnAKe performs best when there are enough iterations for the samples to adequately fill the search space: this is highlighted by poor performance in Perm10D and by ordinary performance in low-budget experiments (see Appendix F, where we include performance comparisons for different budget sizes, $T$).

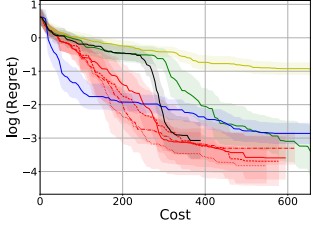 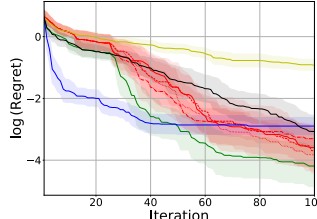 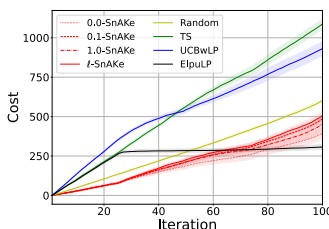

(a) Evolution of regret against input cost. We can see that SnAKe is able to achieve the best regret for low cost.

(b) Evolution of regret with iteration number. SnAKe's final regret is comparable with classic BO methods, and better than EIpuLP.

(c) Evolution of input cost with iteration number. Low cost is achieved by EIpuLP, SnAKe and the TSP-ordered Random optimization.

Figure 3: Results of experiments on the SnAr chemistry Benchmark. SnAKe achieves the best regret out of all low-cost methods. The bounds are created from $\pm$ half the standard deviation of all runs. The best performer, $\ell$-SnAKe, is parameter-free.

## 4.1 Synthetic Functions

**Sequential BO** This section examines the performance of SnAKe against Classical BO algorithms. We compare Expected Improvement (EI) [33], Upper Confidence Bound (UCB) [46], and Probability of Improvement (PI) [25]. We compare against Expected Improvement per Unit Cost [44] (EIpu) as introduced in Eq. (1), with $\gamma = 1$, and with Truncated Expected Improvement [41] (TrEI). We also introduce a simple baseline, where we create a random Sobol sample, and then arrange an ordering by solving the TSP, and never update the path again. We do this for three classical benchmark functions, and six total in the Appendix F.1.1. We set the cost function to be the 2-norm distance between the inputs. Figure 4 shows the results in the top row.

**Asynchronous BO** We explore the asynchronous setting, comparing Local Penalisation with UCB (UCBwLP) [1, 14], Thompson Sampling (TS) [22], and the same Random baseline from the sequential setting. We also test on EIpu with Local Penalisation (EIpuLP), with $\gamma = 1$. The results are shown in last two rows of Figure 4.

## 4.2 Reaction Control on SnAr Benchmark

We test our method on a real-world, SnAr chemistry benchmark [18]. We control four variables; equivalents of pyrrolidine, concentration of 2,4 dinitrofluorobenenze at reactor inlet, reactor temperature, and residence time. We assume changing temperature, concentration and residence time incur an input cost, owing to the response time required for the reactor to reach a new steady state. We assume

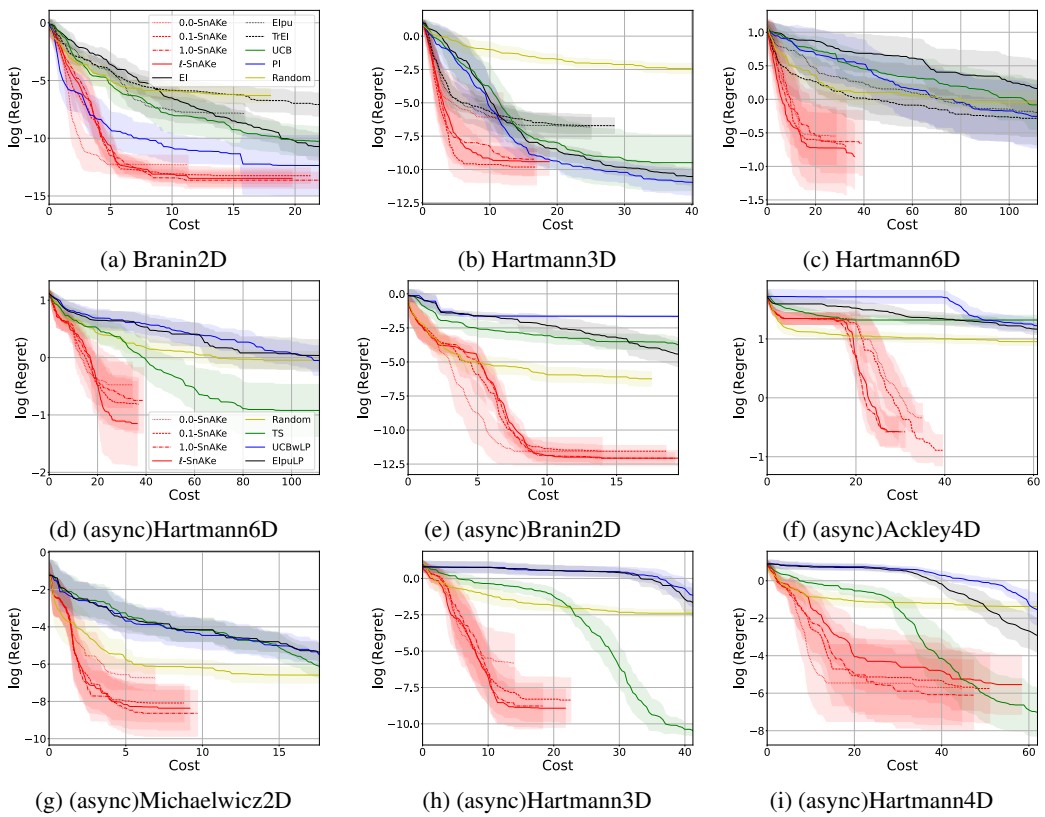

Figure 4: Synthetic experimental results. The first row represents three of the synchronous benchmarks. The last two rows correspond to asynchronous experiments with $t_{delay} = 25$. We plot the average $\log$(regret) achieved against the cost spent. For every experiment, $T = 250$, and we limit the $x$-axis to the maximum cost achieved by SnAKe or Random. $\ell$-SnAKe is the parameter-free version. SnAKe consistently achieves good regret against at cost. The bounds are $\pm$ half the standard deviation of 25 different runs.

the reactor as a first-order dynamic system, where the response to changes in input is given by:

$$(x_s)_i = (x_t)_i + (1 - e^{-s/\alpha_i})(\Delta x_t)_i \tag{3}$$

where $s$ denotes the time after experiment $x_t$ is finished, $(x_t)_i$ denotes the $i$th variable of the $t$th experiment, $(\Delta x_t)_i = (x_{t+1})_i - (x_t)_i$, and $\alpha_i$ is the system time constant. We define quasi-steady state using an absolute residual $\beta_i$, i.e., $|(x_s)_i - (x_{t+1})_i| \leq \beta_i$. For input changes smaller than $\beta_i$, we assume a linear cost, defined by a parameter $\gamma_i$. Combining with the response time from (3):

$$\mathcal{C}_i(x_t, x_{t+1}) = \gamma_i \min\{\beta_i, |\Delta^{(i)} x_t|\} + \max\left\{0, \alpha_i \log\left(|\Delta^{(i)} x_t|/\beta_i\right)\right\} \tag{4}$$

We assume that we can change variables simultaneously, so the total input cost is simply the longest response time within a given set of input changes. That is, $C(x_t, x_{t+1}) = \max_{i \in I_c} C_i(x_t, x_{t+1})$, where $I_c$ is the index set of the control variables. We implement the simulation using the Summit package [10]. More details can be found in Appendix E.7.6. We set a delay of $t_{delay} = 25$, and optimize for $T = 100$ iterations. The results of the experiment can be seen in Figure 3.

## 4.3 Finding Contamination Sources in Ypacarai Lake

As a second real-world example that exhibits costly input changes, we investigate an alternative to the case study introduced by Samaniego et al. [41]. Autonomous boats are used to monitor water quality in Ypacarai Lake in Paraguay. We consider three different objectives, illustrated in Figure 5, created using the Schekel benchmark function (as in [41]). Each objective could correspond to a measure of water quality, such as pH, turbidity, $CO_2$ levels, or many more. We consider the problem

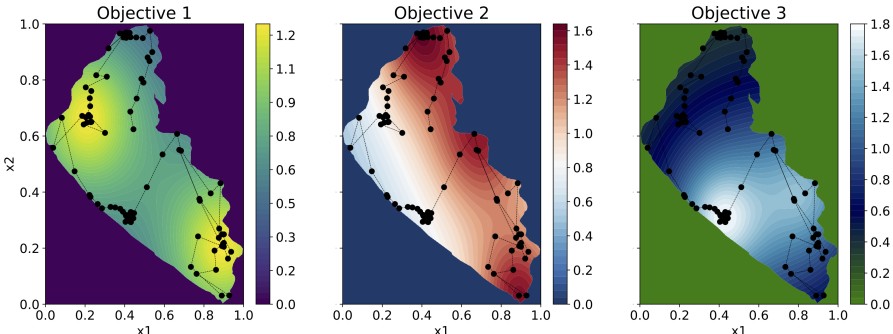

Figure 5: Visualization of Ypacarai Lake, and the objectives we optimize over. We show an example of a SnAKe optimization path on the simultaneous optimization problem (see Appendix C).

Table 1: Ypacarai results. We present the average regret from 10 runs $\pm$ the standard deviation, multiplied by $10^3$. O1, O2, O3 correspond to the different objectives. All runs are terminated after the cost exceeds 10 units (approximately 100 km). SnAKe is the best performer in 2 out of 3 objectives.

|  | Regret O1 | Regret O2 | Regret O3 |
|---|---|---|---|
| TrEI | $0.475 \pm 0.596$ | $0.118 \pm 0.104$ | $0.076 \pm 0.050$ |
| EIpu | $0.249 \pm 0.225$ | $\mathbf{0.068 \pm 0.073}$ | $0.071 \pm 0.049$ |
| SnAKe | $\mathbf{0.074 \pm 0.063}$ | $0.109 \pm 0.186$ | $\mathbf{0.036 \pm 0.048}$ |

of finding the *largest* source of contamination in the lake by maximizing these objectives. The cost between queries relates to the boat physically moving from one place to another. We assume that the boats can travel 10 units, which corresponds to approximately 100km (fuel limitations), and no observation time delay. We run optimization runs on each objective and shows the results in Table 1.

## 5   Conclusion and Discussion

This paper introduces and proposes a solution to the problem of optimizing black-box functions under costs to changes in inputs. We empirically show that the regret achieved by our method is comparable to those of classical Bayesian Optimization methods and we succeed at achieving considerably lower input costs, while being particularly well suited to asynchronous problems. Examples of further work include extending the algorithm so multiple 'SnAKes' can run in parallel, or extending it to the classical multi-objective setting, e.g., using variations of Thompson Sampling [7].

This setting, with input costs penalizing experimental changes, makes a major step towards automating new reaction chemistry discovery, e.g., in line with the vision of Lazzari et al. [27]. We substantially decrease experimental cost with respect to classical black-box optimization, e.g., as used by Fath et al. [9] and McMullen and Jensen [32]. In the real-life SnAR benchmark, SnAKe spends 45-55% of the cost while making similarly strong predictions to classical BO. The synthetic benchmarks and Ypacarai Lake example offer similar advantages.

## Acknowledgments & Disclosure of Funding

JPF is funded by EPSRC through the Modern Statistics and Statistical Machine Learning (StatML) CDT (grant no. EP/S023151/1) and by BASF SE, Ludwigshafen am Rhein. SZ was supported by an Imperial College Hans Rausing PhD Scholarship. The research was funded by Engineering & Physical Sciences Research Council (EPSRC) Fellowships to RM and CT (grant no. EP/P016871/1 and EP/T001577/1). CT also acknowledges support from an Imperial College Research Fellowship.

We would like to thank the reviewers and meta-reviewers throughout the whole peer review process for their time, attention, and ideas, which led to many improvements in the paper. Discussions with colleagues in the Imperial Department of Computing led to the ideas of $\epsilon$-Point Deletion and the nonparametric $\ell$-SnAKe. Linden Schrecker improved our understanding of the chemistry application.

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
