# OpenReview forum: "SnAKe: Bayesian Optimization with Pathwise Exploration"
_NeurIPS.cc/2022/Conference — NeurIPS 2022 Accept_

### Official Review · Reviewer_aedf · 2022-07-09

**Rating:** 6
**Confidence:** 3
**Soundness:** 3 good
**Presentation:** 2 fair
**Contribution:** 3 good

**Summary:**

The paper introduces a new variant of cost-aware BO: It is more expensive to evaluate two points one after the other if they are farther away from each other than if they are close to each other. A method (SnAKe) for this setting is suggest, that approaches the problem in the following way: In each iteration, a new batch is sampled, using Thompson seedling. The evaluation order of the points in the batch is formulated as a TSP and optimized. The method runs asynchronously, i.e. points are resampled after every new observation. However, this leads to too much exploitation, which is why sampled points epsilon-close to already evaluated points are not taken into account. The method shows a good performance on synthetic benchmarks and two real-world examples.

**Questions:**

The optimized paths in Figure 4 all look the same, even though the objectives are different. Is there a reason or could this be a plotting error?

**Limitations:**

The authors address limitations: The method performs less well in higher-dimensional search spaces and low-budget settings.

**Strengths And Weaknesses:**

The work is original, and I think it is an interesting question how to balance exploration with exploitation, when there are additional external factors (like costs), that influence the trade-off between the two. The setting is also signifiant for real-world applications of BO.
The experiments support the claims of the paper. The real-world experiments are nice. A strength of the method is that it seems to be robust to the choice of hyper parameter epsilon.

The writing could be improved in some aspects, e.g.:
* In section 2 (line 68), the abbreviation "EI" is used without definition. It becomes clear from the context that it means "Expected Improvement", but I would mention this somewhere explicitly.
* Maybe one could change the color-scale in Figure 2. It's a bit difficult to see the gradation.
* I would suggest to have Section 3.3. as part of Section 3. One could also mention the definition of regret in Section 3.1
* I am not a native English speaker, but the wording sometimes seems a bit informal to me.

---

> ### Author Response · Authors · 2022-08-01
> **Author's Rebuttal**
>
> Thank you for taking the time to review and provide feedback.
>
> Thank you further for pointing out errors and for suggesting formatting improvements. You can see the changes in the revised paper. We would appreciate some more specific examples of where the writing seems informal.
>
> Questions:
>
> 1. The optimized paths in Figure 4 all look the same, even though the objectives are different. Is there a reason or could this be a plotting error?
>
> By mistake, we used the plots for the simultaneous objective example which is in the Appendix (that is, where we optimize all three objectives in a single run - you can see the path goes through the optimum of all objectives). Thanks for pointing this out, we will correct this.

---

### Official Review · Reviewer_V9c2 · 2022-07-11

**Rating:** 5
**Confidence:** 3
**Soundness:** 2 fair
**Presentation:** 3 good
**Contribution:** 3 good

**Summary:**

This work presents a low-cost Bayesian optimization solution, which is inspired by real-world applications. The proposed method is analyzed mathematically in Section 3. Experimental results in Section 4 shows the effectiveness of the proposed method when compared with several state-of-the-art methods. Comprehensive experimental results are shown in the appendix.

**Questions:**

Why does UCBwLP perform better in Figure 3(b) than the proposed method?

As the proposed method is claimed to be low-cost. Can the authors provide quantitative analysis of this work with other existing methods in terms of cost?

What's the impact on the proposed work if the delay of observations varies?

**Limitations:**

Low-cost analysis of the proposed work is not provided.

**Strengths And Weaknesses:**

The solution has a novel idea of optimizing sample paths while minimizing the input costs. The paper is well-written with understandable examples and mathematical formulation. The comprehensive experimental results support the effectiveness of the work.

However, there are some minor drawbacks: 1) The notation used in formulation can be pre-defined so that readers understand them better; the paper would be self-contained if notation is well-defined. 2) Minor editing flaws: Figure 5 is mentioned before Figure 4; Fonts in figures can be enlarged for better readability.

---

> ### Author Response · Authors · 2022-08-01
> **Author's Rebuttal**
>
> Thank you for taking the time to read and give feedback on the paper. A quick discussion:
>
> 1. We will improve notation and clarity based on all the feedback from the reviews. We would appreciate any specific feedback on the parts where you thought we could do a better job.
>
> 2. We have flipped the order of Figures 4 and 5.
>
> Questions:
>
> 1.Why does UCBwLP perform better in Figure 3(b) than the proposed method?
>
> This Figure plots regret against iteration number. Because UCBwLP is not considering the cost of moving from one experiment to the other, it is quickly jumping around the search space and reaching an optimal area in much fewer iterations - however this comes at a very high cost. SnAKe takes into account the whole budget and avoids unnecessarily large jumps as it knows there is spare budget to explore the rest of the space later. Note in Figure 3(b) that by the end of the optimization SnAKe catches up  and performs very well. Most importantly, as we can see from Figure 3(a), if we take the cost into account, UCBwLP is performing poorly when compared to SnAKe.
>
> 2. As the proposed method is claimed to be low-cost. Can the authors provide quantitative analysis of this work with other existing methods in terms of cost?
>
> Figure 3(c) compares the cost of each method, in this example SnAKe achieves lower cost than all methods except EIpu - however SnAKe achieves significantly better regret than EIpu, suggesting EIpu explores the space too slowly or gets stuck at local minima. In Appendix F, we have a thorough analysis of the final cost achieved by each method. Overall, SnAKe consistently manages to achieve both low cost and low regret.
>
> 3. What's the impact on the proposed work if the delay of observations varies?
>
> To avoid confusion and try to make the method as clear as possible, we assume a fixed delay. However, SnAKe does not assume that the delay is fixed or known: it could be changing in a deterministic fashion or even randomly. One of the advantages of SnAKe is that once a path is determined, it can be followed until the budget is finished - there is no expectation of receiving new observations. However, if there are new observations, we can immediately update the path. Usually, more delay means performance will get worse, however, we had an interesting finding where in the high multi-modal example (Ackley 4D), the delay helped improve performance as it became more difficult to get stuck in local optima.

---

### Official Review · Reviewer_5kVY · 2022-07-11

**Rating:** 5
**Confidence:** 4
**Soundness:** 3 good
**Presentation:** 3 good
**Contribution:** 2 fair

**Summary:**

This work tackles the problem of cost-aware optimization of expensive-to-evaluate functions for which large changes in the input parameters between consecutive queries incurs large cost. The paper assumes the cost function, a function of the current parameters and parameters evaluated immediately before, is known a priori. The paper proposes a novel Bayesian optimization algorithm for this problem setting that plans a path of future designs. The method first generates a batch of new promising designs via Thompson sampling and then orders the points by formulating the problem as traveling salesperson problem (TSP) and approximately solving the TSP problem with simulated annealing. To reduce the issues of sampling points that are close to one another, the paper proposes to generate a larger than needed batch and delete the closest design to each previously evaluated point if the closest design is within a certain radius (a new hyperparameter) of a previously evaluated design--otherwise a random point is deleted. The method works for the asynchronous setting where the next design must be selected before some number of previous designs have returned.

**Questions:**


* Are there other compelling examples/demonstrations that can be included to illustrate other instances of this problem class?
* What is the typical level of parallelism (asynchronicity)?
  * The path ordering issue is relevant when designs are evaluated sequentially (as in the chemistry problem in Figure 6). This point should be highlighted upfront. Typically, in batch/async BO, it is assumed that designs are evaluated simultaneously, but the path ordering problem is only relevant when there is some sequence to the evaluations. Although the lake problem does have sequential evaluation (if you only have one boat), there does not seem to be motivation for asynchronous generation/scheduling, but certainly non-myopia is important. The question then becomes how does SNAKE compare with a non-myopic cost-aware BO method?
* What is the wall time of SNAKE to generating a batch of points? Given that it is expensive, empirical time results should be included in the paper
* The claim that SNAKE can optimize multiple objectives should be toned down. Typically, the goal in multi-objective optimization is to learn trade-offs between objectives, but this paper simply optimizes the sum of the objectives
* Should the number of generated points for purposes of the TSP (non-myopic planning) potentially be different from the level of parallelism? It appears that the number of generated points is currently the level of parallelism, but this also means that the algorithm gets more myopic as the parallelism decreases.
* L185: shouldn’t p_t on the RHS be p_{t_e}? The result doesn’t obviously hold in that case.
* How does using epsilon-point deletion compare penalizing the objective based on the distance to previously evaluated or selected designs? Is the a reason to prefer one approach over another. If not, empirical justification for the choice seems warranted
* L204: “may be approximately constant with time”. Can you elaborate on this? Currently this statement sounds hand-wavy

  * How does the method change if there are multiple queues (for example multiple boats in the lake problem or multiple places to run experiments in the chemistry problem)?
  * The level of parallelism should be stated clearly for all problems
* L223: the adaptive grid is mentioned very briefly, but it warrants further motivation and discussion in the main text. It also appears to introduce new hyperparameters
* 10 replications does not seem sufficient. The error bars of +/- one half of a standard deviation are not informative and raise questions about the soundness of the results
* The clarity of the results section would benefit if the baselines were clearly described upfront. TREI is never defined in the paper. L252 mentions LPwUCB, but figures say UCBwLP. EIpuLP is not defined.
* Why is no other cost-aware non-myopic BO method compared against? For example, Lee et al., 2021.
* L144: I recommend moving Figure 1 up further in the paper. It is mentioned on page 4, but does not appear until page 6
* L151: Why focus on a ball centered at `a` rather than x_{t-1} or the set of balls centered at each previous design (stating `a` is a previously evaluated design or design in the batch suffices)?
* Figure 3: the methods should at least be described before the results

Lee et al., 2021. A Nonmyopic Approach to Cost-Constrained Bayesian Optimization.

**Limitations:**

In L240-241, some limitations of SNAKE are highlighted. Further discussion of SNAKE's limitations would be good. No societal impacts are discussed, but such discussion seems unnecessary.

**Strengths And Weaknesses:**

Strengths
* Optimizing sequences of designs (paths) is a neat idea and as far as I know, a novel formulation.
* The paper is largely well written (aside from some undefined acronyms and minor comments listed below).
* The chemistry problem is a great example

Weaknesses
* However, the prevalence of these problems appears limited, and therefore the contribution appears to be a highly nuanced class of problems (see discussion of sequential nature of evaluation below).
  * The chemistry problem showcased in Figure 6  and included the experiments is a compelling example, but it does not seem that the other real-world inspired example problem (the lake problem ) fits this problem class.
* In addition, the proposed technique for dealing with sampling nearby previously selected designs (of deleting nearby points) is not theoretically grounded.
* The mathematical analysis in Sec 3.7 has weak insights.
* There are no significant theoretical results

---

> ### Author Response · Authors · 2022-08-01
> **Author's Rebuttal**
>
> Thanks for the time to read the paper and provide thorough feedback.
>
> 1. The chemical problem is an incredibly important example. Micro-fluidic technology is heavily researched, with applications to many areas including: COVID (Yin et al, Lab on a Chip, 2022), cancer (Kim et al, Lab on Chip, 2022), genotyping (de Olazarra et al, Lab on Chip, 2022), etc. Further, any Bayesian Optimization procedure requiring optimizing temperature or steady-states would benefit from SnAKe: we are not limited only to the micro-scale but almost any chemical optimization. In revision, we will clarify the importance of this challenge.
>
> 2. There is a misunderstanding about point deletion. We do not replace Thompson Samples with random points. We create a large sample of $T$ points, but we only have a budget of $T-t$ evaluations left, so we must remove $t$ points from the large sample. We remove points from the sample if they are similar to any of the previously $t$ evaluated designs (we do not replace the removed points). After the deterministic removals, it is likely the sample is still larger than our remaining budget, so we balance it by randomly removing points from the sample. We never resort to random sampling.
>
> 3. The Section 3.7 analysis is not rigorous because we order the points using the TSP. The TSP adds an uncommon dependency between the training points and makes the maths very difficult to analyze. We cannot use results from literature as we break classical assumptions. But the insights are not weak. The insights give two key pieces of information: (a) Re-sampling naively has a fatal flaw and (b) Point Deletion allows exploitation for time $T p_t$, which intuitively seems like the correct budget to allocate to a promising area ($p_t$ can be interpreted as the probability that an area contains the global optima).
>
> Questions:
>
> 1. (Parallelism)
> There is no parallelism, we only study the sequential setting (see L95).  Running "multiple SnAKes" in parallel is possible, but this is future work. By "asynchronicity" we mean the difference in time between making decisions and receiving observations. SnAKe is completely non-myopic: it plans ahead for the whole budget.
>
> 2. (Wall-time)
> Wilson et al. [2021] make creating the samples from the GP quick. We then need to optimize $T$ functions. The wall-time depends on the optimizer, but if is has complexity $B$, then the run-time is $BT$. For $T = 250$, an iteration of SnAKe had similar wall-time as Local Penalization. Generating the batch is highly parallelizable as all points are independent, so SnAKe could be sped up significantly.
>
> 3. (Multiple objectives)
> There may be some confusion in the multiple objective experiment. We do not optimize a sum of objectives, we simultaneously optimize independent objectives. We do this by creating the large batch of points using a mix of Thompson Samples from different models (ratio 1:1:1). We agree this example is not fully relevant to our paper's aim. Additionally, we are not using one of the common MO optimization algorithms. We will tone down the example by only including it in the appendix.
>
> 4. (L185-204 issues)
> The results still hold as we are assuming that $p_t \rightarrow p \neq 0$ 'quickly' (as we sample points in $A$), which was not made clear in the paper (hence approximately constant line). So the equation becomes $(1-p)^{T'-t} = (1-p)^{T-{t_e}}$. Using balls centered at $x_{t-1}$ changes the area at each time-step, which adds complexity.
>
> 5. (Point deletion vs Penalizing Objective)
> Local Penalization (LP) penalizes based on distance between points, but this is usually done with acquisition functions. To our knowledge, there is no evidence that penalization works well on GP samples. Further, LP is used on points being evaluated - this limits the number of penalizers, for example in our case when $T = 250$ we would have 249 local penalizers in the last iteration, which seems problematic. Balancing the strength of the penalization so that we exploit promising areas enough but don't get stuck is a further issue.
>
> 6. (Adaptive grid)
> Unfortunately due to space constraints we moved the main motivation to the appendix E.1. We can reintroduce it in the camera ready version.
>
> 7. (Number of replications)
> We ran more experiments and increased from 10 to 25 replications. The new results are in the revised paper. The conclusions remain the same.
>
> 8. (Lee et al., 2021)
> We reference Lee et al. [2021] in the related work section, but we believe it does not apply to the computations because it can only look-ahead for short time horizons (up to 4 in the paper) and there is no obvious asynchronous extension. Lee et al.  [2021] requires model updates, otherwise it will lead to redundant sampling. The acquisition function, based on roll-out, is unusual, so it is not clear fantasizing or local penalization can be applied.
>
> 9. (Clarity of paper)
> Thanks for pointing out the typos and formatting issues.

---

> > ### Comment · Reviewer_5kVY · 2022-08-05
> > **response**
> >
> > Thanks for the detailed response.
> >
> > > There is a misunderstanding about point deletion.
> >
> > Thanks for the clarification. Not sure how I missed that. I have updated my review.
> >
> > > (Parallelism) There is no parallelism, we only study the sequential setting (see L95).
> >
> > A multi-snake variant seems quite practical. It might be worth adding this in the discussion.
> >
> > > We do not optimize a sum of objectives, we simultaneously optimize independent objectives. We do this by creating the large batch of points using a mix of Thompson Samples from different models (ratio 1:1:1).
> >
> > Got it. This should be made clearer in the main text. To a reader who is accustomed to the multi-objective in the tradition sense,  "For the experiment we use the ratio 1:1:1. For TrEI and EIpu, we adapt to the multi-objective case by optimizing the first objective until we have travelled 3.3 units (approximately 33km), after which we change to the subsequent objective. " sounds a lot like some kind of scalarization is optimized with equal weights. I would make it clear in the main text that this is simultaneously, independently optimizing multiple objectives --- not to learn trade-offs, but to learn the optimal design for each objective individually.
> >
> > > (L185-204 issues) The results still hold
> >
> > Thanks for updating this. I am still perplexed. "However, the question then becomes, how long will it take us to
> > leave?"
> >
> > The only reason that full escape probability does not improve with a larger budget is because the formulation focuses on fully escaping (probability of all points escaping), rather the number of escaping points. Clearly the expectation of the latter increases with the budget. Why is fully escaping the criteria of interest rather than number of escapes?
> >
> > > (Adaptive grid) Unfortunately due to space constraints we moved the main motivation to the appendix E.1. We can reintroduce it in the camera ready version.
> >
> > There is some connection between the issue of closeness to previously evaluated designs and the choice of grid points in the adaptive grid. It would be nice to have a discussion of why you don't just choose a grid to mitigate closeness -- presumably because at some point exploitation around the best point is the prudent thing to do.
> >
> > > (Number of replications)
> > Thanks
> >
> > I have updated my score.

---

> > > ### Comment · Reviewer_5kVY · 2022-08-05
> > > **follow-up question**
> > >
> > > Say several points in the batch are nearby one single previously evaluated design (and further from the other previous evaluated designs). the epsilon deletion method will remove the closest point, but the rest may not be deleted. Is there a way that your method can guard against this potential failure mode?

---

> > > > ### Author Response · Authors · 2022-08-05
> > > > **Response**
> > > >
> > > > We do not consider this a failure mode, but a feature of Point Deletion. If there are several points inside an epsilon ball around, say $x_t$, and further from the other previous evaluated designs, we take this as evidence that the region is worth exploiting further (as we have only sampled there once). Only deleting one of the points is a good thing; it allows SnAKe to keep sampling this area until there is a balance between the number of previously evaluated designs and the number of samples in the area - and then we will finally continue exploring the search space.
> > > >
> > > > This is important because we want to avoid the need to frequently come back to promising areas, since moving around the search space incurs input costs.

---

> > > ### Author Response · Authors · 2022-08-05
> > > **Response**
> > >
> > > Thank you reading our rebuttal carefully and answering in such detail.
> > >
> > > Adding parallel extensions to the discussion and clarifying the multi-objective example in the main text are great suggestions, we will make the changes.
> > >
> > > > Why is fully escaping the criteria of interest rather than number of escapes?
> > >
> > > This is a good point, after giving it some thought we came to the conclusion that we are implicitly assuming the worst-case scenario for convergence, which is that we resample at every iteration:
> > >
> > > Assume $x_{t}$ is in a promising area, $A$. If we fully escape $A$, then we guarantee that $x_{t+1}$ will have escaped. On the other hand, if there is at least one sample in $A$, then it is likely that the TSP will choose  a sample in $A$ to be $x_{t+1}$. This would not really be a problem if we do not resample again, since we could simply evaluate however many non-escapes we had in the sample and then leave, but the problem is the algorithm will keep resampling to update the path - and each time we resample we are reintroducing exploitative points in $A$! So fully escaping is the only way to guarantee the algorithm will not get stuck in an area. Because of the constant resampling, if there is even just a single non-escape at every iteration we will remain stuck (hence the focus on fully escaping rather than number of escapes).
> > >
> > > This implicit assumption, that we resample at every iteration, will be stated in the main paper explicitly. We will also add that this assumption should be common in practice: after possibly an initial short time where we do not receive any data (due to delays), we expect a constant stream of observations and hence we should be constantly resampling to update the path. Indeed, this was the case in all our experiments.
> > >
> > > > There is some connection between the issue of closeness to previously evaluated designs and the choice of grid points in the adaptive grid. It would be nice to have a discussion of why you don't just choose a grid to mitigate closeness -- presumably because at some point exploitation around the best point is the prudent thing to do.
> > >
> > > This is another great point and we agree, it is important to have the flexibility to exploit near the best point observed so far. A second issue: if we create all samples with the same grid, there is a chance multiple samples will fall in the same grid point and worsen the redundant sampling problem. Alternatively, we could remove grid-points once a sample falls on them, however, this means that we would need to sample sequentially instead of in parallel, losing the main advantage of using Thompson Sampling. Finally, sampling on a fine enough grid is prohibitively expensive in high dimensions. We will add this discussion to the justification of the adaptive grid, making it clear that it helps us control the computational expense of the TSP but not solve the convergence problems that arise from resampling.

---

### Official Review · Reviewer_UNKt · 2022-07-28

**Rating:** 4
**Confidence:** 3
**Soundness:** 2 fair
**Presentation:** 2 fair
**Contribution:** 2 fair

**Summary:**

The paper proposes an efficient algorithm for a novel optimization task where the distance of candidates to be evaluated incurs input costs. The algorithm penalizes the input variations on top of the traditional Bayesian Optimization algorithm while leveraging the batched Thompson sampling to diverge the candidate picks. Traveling salesman The problem is solved to create an efficient path for the batched query.

**Questions:**

1. For figure 3, does 3.a and 3.b share the legend in 3.c? If so, I would suggest putting the legend in a shared space instead.
2. In equation (1), the \gamma is mostly set to be 1 and could be problematic. For example, when the search space X is scaled up, the shape of the objective function remains the same while the behavior of the algorithm could change as the cost(x, x_{t-1}) is scaled up. Is there any normalization of X in the preprocessing? Could the author demonstrate the corresponding performance of different \gamma?

**Limitations:**

The author addresses the concerns.

**Strengths And Weaknesses:**

- ***Strengths***
1.  The paper studies a novel distance-aware optimization setting motivated by real-world application.
2.  The paper proposes a novel solution to penalize the input variations while avoiding being stuck in local areas in the optimization by the combination of batched Thompson sampling and the proposed $\epsilon$-point deletion. The algorithm design is motivated by insights derived from the analysis of the desired escape from the local optimum.
3. The algorithm empirically validated that the input cost dropped.

- ***weaknesses***
1.  The cost in this problem setting is not defined in a principled way. The paper attempts to minimize the regret within a certain number of iterations while minimizing the input variation cost. Essentially there is one optimization objective and two types of costs considered. One cost is the number of iterations, which is the typical BO cost given each evaluation is expensive, and the other cost is the input variation. The triplet structure of the problem incurs a challenge in performance measurement. Instead of merely comparing regret within a certain iteration as in the traditional BO setting, the triplet requires the additional comparison on costs, which make the performance of different algorithms less comparable.
For example, as shown in figure 3, EIpuLP achieves lower cost while bears higher regret compared to the proposed algorithm, it becomes unclear whether EIpuLP is better in figure 3. Similarly, when the same mixed performance appears for different hyperparameter settings, there lacks a principled way to pick the better hyperparameter. Unless there is a weighted sum of the two different costs where the weights reflect the importance of these costs in the application, it's hard to compare algorithms and hyperparameter choices. The corresponding algorithm design should also consider the weights. Otherwise, it's possible that there exists an algorithm that is significantly superior on one cost while lags on the other. Algorithms like Genetic algorithms that only conduct small variations in each step could potentially bear much lower input costs while incurring a much larger amount of iteration.
2. As mentioned, Genetic algorithms could be included in the baselines.
3. The paper argues that the $\epsilon$-Point Deletion is robust to the hyperparameter choices and proposes to leverage the lengthscale of the kernel to adaptively pick the value of $\epsilon$. The robustness is counter-intuitive as the local (sub)gradients of the objective function could be different even assuming a global smoothness guarantee for the objective function. Additional study on the $\epsilon$-point deletion algorithm behavior might be necessary to explain the robustness, especially on the ratio of the random deletion among all deletions for each performance curve.
4. The confidence intervals of the regret curves for experiments are large and potentially indicate the lack of statistical significance of the results. Could the author provide corresponding p-values?

---

> ### Author Response · Authors · 2022-08-01
> **Author's Rebuttal**
>
> Thank you for taking the time to review and provide feedback. There are a few points we would like to bring up:
>
> 1. We do not believe the problem is ill defined or that there are two types of costs considered. We do not consider the cost of the number of iterations, instead we assume there is a fixed budget of experiments to run. All algorithms select exactly $T$ experiments, so the iteration-cost is not important. This is common in chemistry where the amount of chemicals bought, and hence the number of experiments, is fixed. This is why we mainly use regret-cost graphs to compare the different methods. For example, in Figure 3(a), shows that even though EIpuLP achieves a final lower cost, SnAKe is outperforming the regret of EIpuLP for the same cost. Overall, we believe that we have empirically shown that in most cases SnAKe achieves lower cost without having to sacrifice performance in terms of regret. Further, the regret per iteration for SnAKe is not very meaningful, because SnAKe uses the number of available experiments to modify the speed at which it traverses the search space - if we had a larger budget it would move slower, but if we had a smaller budget it would move faster. In the Appendix F we explore the performance of SnAKe for different budgets.
>
> 2. Genetic Algorithms are not well suited for this problem a few reasons: (a) Evaluating the black-box function is expensive, and GA algorithms are not sample efficient enough for this setting (or any Bayesian Optimization setting) (b) We are assuming we carry out experiments sequentially, so there each generation only have 1 individual (c) We have asynchronicity to deal with.
>
> 3. The robustness to $\epsilon$ is intuitive for one reason - eventually all excess exploitation will be centered around a very very small area (where the best observed local optima lies), so the size of the epsilon balls is non-important as long as it is non-zero. For further evidence, consider Nava et al. [2022, AISTATS] who explore Thompson Sampling's batch diversity and show it can be too exploitative with local optima.
>
> 4. To ensure more confidence in the results, we added 15 runs per experiment (can be seen in the revised paper). The conclusions remain the same.
>
> Questions
>
> 1. For figure 3, does 3.a and 3.b share the legend in 3.c? If so, I would suggest putting the legend in a shared space instead.
>
> Thanks for the suggestion, we agree that a shared legend would look better. We we unable to fit the change due to limited space, but we can fit it for the camera ready version.
>
> 2. In equation (1), the $\gamma$ is mostly set to be 1 and could be problematic. For example, when the search space X is scaled up, the shape of the objective function remains the same while the behavior of the algorithm could change as the cost($x$, $x_{t-1}$) is scaled up. Is there any normalization of X in the preprocessing? Could the author demonstrate the corresponding performance of different $\gamma$?
>
> We agree that the method will be sensitive to the choice of $\gamma$! This highlights why traditional cost-aware literature, in its current form, is not suited for our specific problem. We do not believe there is an obvious way of choosing this parameter, so we set $\gamma = 1$ as the baseline, however, it not the purpose of this paper to find a way of tuning this hyper-parameter.

---

> > ### Comment · Reviewer_UNKt · 2022-08-09
> > **Reply to Author's Rebuttal**
> >
> > I appreciate the detailed responses by the authors. I believe the change in experiments and illustrations addresses some of my concerns. While I confirmed my concerns on the problem definition.
> >
> > - As the author mentioned, evaluation of the function is expensive, that's why the number of iterations the algorithm used is considered a cost. The author suggested that in some real-world applications the total budget in terms of available iteration T is fixed. However, from the perspective of algorithm design, the algorithm should be adaptive to different T, meaning the performance metric of an algorithm should take T into account. Bayesian optimization algorithms typically aim to minimize regret with respect to the number of iterations due to the assumed expensive evaluation. And the regret bounds of BO normally include T. If the author wants to argue the proposed algorithm efficiently tackles the discussed distance-aware variant of the Bayesian optimization problem, the regrets against iterations (or more generally defined cost) is a necessary performance metric.  In this sense, as I mentioned in the review, there are actually two acting costs and two different performance metrics (regret against iteration and regret against input cost) correspondingly in the paper. The problem definition makes a fair performance comparison between different algorithms/hyperparameters impossible.
> >
> > - Question 2 in the original review also gives rise to the need for a uniform performance metric.  Effectively the shape of the unknown objective function decides the optimal optimization trajectory that minimizes the regret while doesn’t incur a relatively large distance-aware cost. Therefore the optimal optimization trajectory should be invariant to scaling. The proposed algorithm has such property and is actually scaling invariant when choosing small $\epsilon$. **However**, the current input cost is not defined in a relatively stable rank compared to the number of iterations. Therefore when the search space is scaled up by a certain constant, the input cost is also scaled up and tends to discourage exploration. Then reducing the exploration and more exploitation could naturally strike a better balance between regret and cost.  This is certainly not desired for a sound problem definition.
> >
> > In summary, I feel like the proposed algorithm could potentially bring improvement in the discussed distance-aware setting compared to existing cost-aware BO algorithms or genetic algorithms. The work needs to provide more principled performance metrics and show the empirical results accordingly rather than relying on the current unprincipled comparisons.

---

> > > ### Author Response · Authors · 2022-08-09
> > > **Author Response**
> > >
> > > We would like to thank you for reading the rebuttal and providing a detailed response.
> > >
> > > 1. The problem as defined in the paper can be reduced to: “For a given budget, $T$, optimize the black-box function $f$ for the cheapest cost (e.g. distance between variables) possible”. As mentioned in the rebuttal, this setting is inspired by real-world chemistry applications where the number of possible experiments is deterministic. We believe this problem is well defined.
> > >
> > >
> > >    Because we are focusing on planning-ahead, and on being non-myopic, knowing the budget is a requirement to be able to obtain a good solution with respect to the distance metric.
> > >
> > >    Nonetheless, we agree that the performance with respect to the budget is also important, which is why we tested different budgets as part of our experiments. The results can be found in Appendix F, and they are mentioned in the limitations: SnAKe performs best when the budget is large, and worse when the budget is small (see L239-L241).
> > >
> > >
> > > 2. We would like to highlight that the way that we define the problem is for general cost functions, not just distance. Therefore, the cost-metric should be defined by the real-world application and this should consider any scaling required. For example, in the SnAr experiment we do not use distance as a cost function, but a real-world inspired criterion.
> > >
> > >    Also, the way the problem is defined is scale-invariant: “For a given budget, $T$, optimize the black-box function $f$ for the cheapest cost (e.g. distance between variables) possible.”. As the reviewer points out, SnAKe is scale-invariant, and tackles this problem directly. It is other cost-aware methods whose behavior will change with different scaling and this might lead to issues: for example, if exploration is discouraged, it might not optimize the function well.

---

### Meta-Review · Area_Chair_pyzr · 2022-08-23

**Recommendation:** Accept
**Confidence:** Less certain

**Metareview:**

There was disagreement among the reviewers about whether this paper should be accepted. But taking the long view about how this paper might be perceived in 10 years, and reading the paper in detail myself, I lean towards acceptance being the right decision for this paper. My reasoning:
- The problem being solved is completely novel as far as I am aware, and is well-motivated by a realistic real-world scenario in a large field of research.
- The proposed approach is a highly-original variation on BO that, despite not being justified by theory at all, seems like a plausible direction to explore to get faster algorithms.
In light of the above, the potential impact of the paper might be large: we may discover other applications that have the same cost setup, we may encourage ML researchers to take on chemistry-motivated problems, we may encourage theory researchers to analyze the method (or more likely to come up with methods that are justified), or we may see empiricists use similar ideas in other settings.

That being said, the reviewers point out some completely-valid criticisms. So I expect to see quite a few updates in the final version of the paper. Please comb through the reviews carefully (particularly the two critical reviews) and update the paper based on the reviewer's comments, including and beyond what is already written in the author response. I could imagine many readers having similar issues (e.g., tone down any indication that the algorithm is theoretically justified from a regret perspective), and some of the comments represent ways the paper could be complete in its exploration of the topic.

**Award:**

No

---

### Decision · Program_Chairs · 2022-09-14

Accept